# Design of Switched-Capacitor DC-DC Voltage-Down Converters Driven by Highly Resistive Energy Transducer

**Yosuke Demura and Toru Tanzawa ***

Faculty of Engineering, Shizuoka University, Hamamatsu 432-8561, Japan; demura.yosuke.18@shizuoka.ac.jp
* Correspondence: toru.tanzawa@shizuoka.ac.jp

**Abstract:** Electrostatic vibration energy transducers have a relatively high output impedance ($R_{ET}$) and open-circuit voltage ($V_{IN}$), so that voltage-down conversion is required for sensor/RF ICs. Switched-capacitor converters are the best candidate to create small-form-factor technology and are a low-cost solution because of their capability to fully integrate into sensor/RF ICs. To design switched-capacitor voltage-down converters (SC-VDCs) with a minimum circuit area for electrostatic vibration energy transducers, two steps are required. The first step requires an optimum design of DC-DC SC-VDCs driven by high $R_{ET}$ with a minimum circuit area, and the second step requires an optimum design of AC-DC SC-VDCs based on the first step, to minimize the converter circuit area. This paper discusses circuit analysis and design optimization aimed at the first step. Switching frequency, the number of stages and the capacitance per stage were determined as a function of $R_{ET}$, $V_{IN}$ and the output voltage (Vo) and current (Io) to the load, to achieve a minimum circuit area. The relationship between Io and the power conversion efficiency was studied as well. The performance was validated by SPICE simulation in 250 nm BCD technology. An optimum design flow was proposed to design DC-DC SC-VDCs driven by high $R_{ET}$ with a minimum circuit area under conditions where $R_{ET}$, $V_{IN}$, Vo and Io were given. The second design step remains as future work.

**Keywords:** switched-capacitor converter; voltage-down converter; energy transducer; optimum design; electrostatic vibration energy harvesting; fully integrated; IoT

## 1. Introduction

Energy harvesting (EH) is technology for harvesting power for IoT edge devices from environmental energy using energy transducers (ETs) [1]. Electrostatic energy transducers (ES-ETs) can convert vibration energy into electronic power [2,3]. Due to high output impedance ($R_{ET}$), open-circuit voltages ($V_{IN}$) have to go beyond 10 V to generate power of 10 μW or larger. Switching regulators were proposed in [4,5] with a high-voltage full-bridge rectifier. An HV rectifier is composed of four diodes for converting the AC power of ES-ETs into DC power in the converter. As the DC voltage is much higher than the maximum voltage acceptable in sensor CMOS ICs, power management circuits in DC-DC converters need to be fabricated using a BCD process, which provides an HV CMOS operating even at high voltages of 10 V or higher. Buck converters require external components, such as inductors, capacitors and resistors (LCRs), to convert the DC-input voltage of an order of 10 V into an output voltage of an order of 1 V. DC-DC buck converters are used in applications of very high-power conversion. In [6], monolithic integrated high frequency GaN DC-DC buck converters were proposed to output 15 V at a power density of 1 W/mm². Another type of switching converter is a boost converter. In [7], a boost converter with a piezoelectric energy harvester could generate 1 V from a low-input voltage of 0.12 V with an output power of 4.2 mW for wearable biomedical applications.

An alternative design for ES-ETs is a shunt regulator, which enables the elimination of inductors [8]. The circuit can sufficiently reduce overstress, even with a standard 1 V CMOS, resulting in full integration, apart from the decoupling caps, and provides a low-cost solution. A drawback of the shunt regulator is low power efficiency. As the peak

open-circuit voltage increases, the power conversion efficiency decreases. A third option is switched-capacitor converters, which can be fully integrated as well, and have moderate power efficiency [9]. In [10], the design of switched-capacitor voltage-up converters for a DC-energy transducer was discussed, where the operating clock frequency was assumed to be constant, regardless of the number of capacitors and the capacitance of each capacitor. To the best knowledge of the authors, there has been no formulation to design switched-capacitor voltage-down converters for highly resistive energy transducers. To optimally design switched-capacitor voltage-down converters (SC-VDCs) for ES-ETs, the first step requires an optimum design of DC-DC SC-VDCs driven by high $R_{ET}$, and the second step requires an optimum design of AC-DC SC-VDCs based on the first step.

This paper discusses circuit analysis and design optimization aimed at the first step. Switching frequency, the number of stages and the capacitance per stage were determined as a function of $R_{ET}$, $V_{IN}$ and the output voltage (Vo) and current (Io) to the load, to achieve a minimum circuit area. The relationship between Io and the power conversion efficiency was studied as well. The performance was validated by SPICE simulation in 250 nm BCD technology. The second design step remains as future work. This paper is organized as follows: Section 2 develops circuit models with no $R_{ET}$ case as ideal and a high $R_{ET}$ condition. The sensitivity of Io on the design parameters are discussed. Optimum clock cycle time and the optimum number of capacitors are determined to maximize Io at Vo. Optimization design flow is proposed in Section 3. The results are also shown.

## 2. Circuit Model

### 2.1. Ideal Case with No $R_{ET}$

Figure 1 illustrates a block diagram of an energy transducer (ET) and a switched-capacitor voltage-down converter (SC-VDC). The electrical characteristics can be expressed by an open-circuit voltage $V_{IN}$ and output resistance $R_{ET}$. In this paper, $V_{IN}$ is assumed to be DC to propose design optimization of SC-VDC driven by a highly resistive ET, which will be able to apply to AC-DC SC-VDCs for electrostatic vibration energy transducers in future work. An SC-VDC is composed of multiple capacitors and switches to vary the configuration of capacitors between input and output terminals in two states per cycle of an input clock CLK.

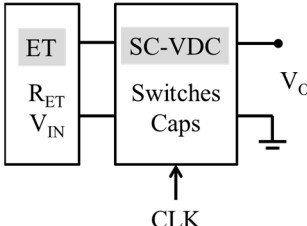

**Figure 1.** Block diagram of energy transducer (ET) and switched-capacitor voltage-down converter (SC-VDC).

Figure 2 shows those two configurations of DC-DC SC-VDC with CLK = H and L in (a) and (b), respectively, namely in a serial state and in a parallel state. An SC-VDC has N capacitors, each of which has the same capacitance C. The input current $I_{IN}$ flows in the serial state. The output currents $I_{OP}$ and $I_{OS}$ flow in the serial and the parallel states, respectively. Figure 2c shows the waveform of $V_P$ at the interface between ET and SC-VDC. The serial state required a longer period (Ts) than the parallel one required ($T_P$) because the capacitors were charged via a large resistor $R_{ET}$ in Ts, whereas they were discharged via a small on-resistance of switches in $T_P$. Thus, the clock frequency was limited by $T_S$. When $T_S$ increased, the amount of charge stored in the capacitors increased, and the cycle time also increased. Overly long $T_S$ could reduce the average output current ($I_O$) because the amount of charge saturated for long $T_S$. On the other hand, when Ts decreased too much, the charge transferred into the capacitors decreased as well. $T_S$ that was too short could

also reduce $I_O$. As a result, there should be an optimum $T_S$ to maximize Io between two extreme conditions. Firstly, we will look at the circuit behavior in the case where $R_{ET}$ is sufficiently small, and then, we will investigate the case where $R_{ET}$ is significantly large.

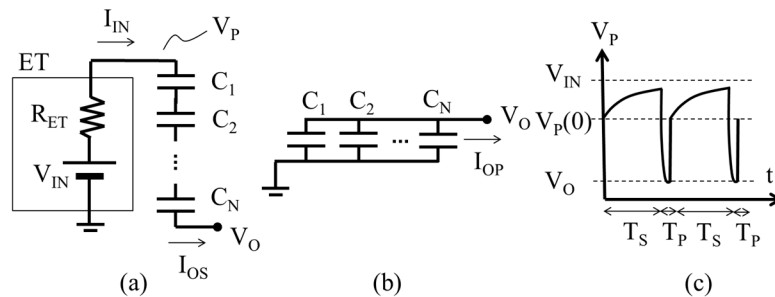

**Figure 2.** Two states of SC-VDC: (**a**) serial connection to extract power from ET; (**b**) parallel connection to output power to $V_O$; (**c**) waveform of $V_P$.

An ideal circuit model of SC voltage-up converters was discussed in [11,12] and the optimum number of capacitors to minimize the circuit area under the condition that the circuit outputs a target current at a given output voltage was also discussed. In this section, we will start with a similar set of equations to represent the circuit performance. All of the parasitic resistance of the power supply and switches was assumed to be adequately small. Therefore, the clock cycle $T$ was considered to be long enough for the charge to be transferred to the capacitors or to the output. Under such conditions, the output charge in the serial state ($Q_{OS}$) and the average current over $T$ ($I_{OS}$) were given by (1) and (2), respectively, as follows:

$$Q_{OS} = \frac{C}{N}(V_{IN} - (N+1)V_o \tag{1}$$

$$I_{OS} = \frac{Q_{OS}}{T} = \frac{1}{N} \times \frac{CV_{IN}}{T}\left(1 - \frac{(N+1)V_o}{V_{IN}}\right) \tag{2}$$

Similarly, the output charge in the parallel state ($Q_{OP}$) and the average current over $T$ ($I_{OP}$) were given by (3) and (4), respectively. The total average output current ($I_O$) was therefore given by (5).

$$Q_{OP} = N \times Q_{OS} = C(V_{IN} - (N+1)V_o) \tag{3}$$

$$I_{OP} = \frac{Q_{OP}}{T} = \frac{CV_{IN}}{T}\left(1 - \frac{(N+1)V_o}{V_{IN}}\right) \tag{4}$$

$$I_O = I_{OS} + I_{OP} = \frac{N+1}{N} \cdot \frac{CV_{IN}}{T}\left(1 - \frac{(N+1)V_o}{V_{IN}}\right) \tag{5}$$

The average input current ($I_{IN}$) was equal to $I_{OP}$ (6). Thus, the input and output power were expressed by (7) and (8), respectively. The power conversion efficiency ($\eta$) defined by $Po/P_{IN}$ was calculated as (9).

$$I_{IN} = I_{OS} \tag{6}$$

$$P_{IN} = V_{IN}I_{IN} = \frac{CV_{IN}2}{NT}\left(1 - \frac{(N+1)V_o}{V_{IN}}\right) \tag{7}$$

$$P_O = V_O I_O = \frac{(N+1)CV_{IN}V_O}{NT}\left(1 - \frac{(N+1)V_o}{V_{IN}}\right) \tag{8}$$

$$\eta = \frac{P_O}{P_{IN}} = \frac{(N+1)V_O}{V_{IN}} \tag{9}$$

$N$ needed to meet (10) to have finite $I_O$ based on (5). From the point of view of high power-conversion efficiency, a larger $N$ was recommended. The largest number of $N$ to meet the equation of $N < V_{IN}/V_O - 1$ had the highest $\eta$. On the other hand, from the point

of view of high $I_O$, a smaller $N$ was recommended because $I_O$ was a monotonic function of $N$, as shown by (11).

$$N < \frac{V_{IN}}{V_O} - 1 \tag{10}$$

$$\frac{\partial I_O}{\partial N} \propto -N\left(\frac{V_{IN}}{V_O} + 2\right) - 2\left(\frac{V_{IN}}{V_O} - 1\right) \tag{11}$$

*2.2. Practical Case with Large $R_{ET}$*

Like the previous model, where the switch resistance was low enough under the slow-switching limit, it was assumed that the parallel state did not require a long period to transfer all of the charges via the switches with low on-resistance. On the other hand, it was assumed that the charges transferred during the serial state were limited by *Ts*. The voltage at the top plate of the top capacitor ($V_P$) in the case where the bottom plate of the bottom capacitor was connected to the output terminal was determined by a differential Equation (12).

$$\frac{V_{IN} - V_p(t)}{R_{ET}} = \frac{C}{N}\frac{d}{dt}(V_p(t) - V_o) \tag{12}$$

With the initial condition as shown in (13), (12) was solved to be (14), where the time constant $\tau$ was given by (15).

$$V_p(0) = (N+1)\,V_O \tag{13}$$

$$V_p(t) = V_{IN}\left(1 - \left(1 - \frac{(N+1)V_o}{V_{IN}}\right) \cdot e^{-\frac{t}{\tau}}\right) \tag{14}$$

$$\tau = \frac{CR_{ET}}{N} \tag{15}$$

The average output current during the serial state over one period of $T = T_P + T_S$ ($I_{OS}$), which was equivalent to the average input current, could be estimated using (16).

$$I_{OS} = I_{IN} = \frac{C(V_P(T_S) - V_P(0))}{NT} = \frac{CV_{IN}}{NT}\left(1 - \frac{(N+1)V_o}{V_{IN}}\right)\left(1 - e^{-\frac{T_S}{\tau}}\right) \tag{16}$$

Because the total transferred charges from the energy transducer in the serial state appeared to be all $N$ capacitors, the charges to the output terminal in the parallel state were given by (17).

$$I_{OP} = NI_{OS} \tag{17}$$

Therefore, an average output current in a period could be estimated by (18). The input and output power were simply given by (19) and (20), respectively.

$$I_O = I_{OS} + I_{OP} = \frac{N+1}{N} \cdot \frac{CV_{IN}}{T}\left(1 - \frac{(N+1)V_o}{V_{IN}}\right)\left(1 - e^{-\frac{T_S}{\tau}}\right) \tag{18}$$

$$P_{IN} = V_{IN}I_{IN} = \frac{CV_{IN}2}{NT}\left(1 - \frac{(N+1)V_o}{V_{IN}}\right)\left(1 - e^{-\frac{T_S}{\tau}}\right) \tag{19}$$

$$P_O = V_O I_O = \frac{(N+1)CV_{IN}V_O}{NT}\left(1 - \frac{(N+1)V_o}{V_{IN}}\right)\left(1 - e^{-\frac{T_S}{\tau}}\right) \tag{20}$$

As a result, when all of the parasitic capacitance, such as bottom and top plate capacitance and junction capacitance of switches was negligibly small, the power conversion efficiency ($\eta$) became (21), which was equal to (9). Equations (16), (18)–(20) became identical to (2), (5), (7) and (8) when $R_{ET}$ approached zero.

$$\eta = \frac{P_O}{P_{IN}} = \frac{(N+1)\,V_O}{V_{IN}} \tag{21}$$

One can find an optimum *Ts* to maximize Io based on (18). With $\frac{\partial I_O}{\partial T_S} = 0$, (22a) held in the case that $\sqrt{\tau T_P} >> T_P$. Similarly, with $\frac{\partial I_O}{\partial N} = 0$ and (22a), (22b) held. As a result, $I_O$ approached the maximum attainable current $I_{O\_ATT}$ given by (22c), which was $I_O$ under impedance matching.

$$T_{S\_OPT} = \sqrt{\tau T_P} \tag{22a}$$

$$N_{OPT} = \frac{V_{IN}}{2V_O} - 1 \tag{22b}$$

$$I_{O\_ATT} = \frac{V_{IN}^2}{4R_{ET}V_O} \tag{22c}$$

### 2.3. Characteristics of SC-VDC for Highly Resistive ET

To see how Io varied as a function of N, C and Ts, a demonstration was performed with the default parameters shown in Table 1.

**Table 1.** Design parameters used as a demonstration.

| Parameters | Default Value |
| --- | --- |
| $V_{IN}$ | 10 V |
| $R_{ET}$ | 100 kΩ<br>(10 Ω for reference) |
| $V_O$ | 1.0 V |
| C | 1.0 nF |
| N | 4 |
| $T_S$ | 10 μs |
| $T_P$ | 100 ns |

When N, C or Ts was varied, the remaining parameters were set at the default values. $R_{ET}$ of 100 kΩ and 10 Ω were used to verify the significance of a large $R_{ET}$ value. In Figure 3a, the number of capacitors (N) was varied. As predicted, Io was maximized with N of one when $R_{ET}$ was sufficiently small. Conversely, Io was maximized to three or four when $R_{ET}$ was quite large, as predicted by (22b). $N_{OPT}$ was estimated to be four when $V_{IN}$ = 10 V and Vo = 1 V by (22b). Figure 3b shows that the response of C to Io was scaled by $R_{ET}$. The ratio of 100 kΩ to 10 Ω was $10^4$. When one drew the curve of $I_O - C$ for $R_{ET}$ = 10 Ω by shifting four orders in the horizontal and the vertical axes, the two curves were matched well. Figure 3c shows that there was an optimum Ts depending on the value of $R_{ET}$. Ts that was too short did not allow charges to be transferred from ET, whereas Ts that was too long simply decreased Io~Qo/Ts, wherein Qo was saturated for long Ts. The estimate equation (22a) gave us $T_{S\_OPT}$ of 15 μs and 1.5 μs for $R_{ET}$ of 100 kΩ and 10 Ω, respectively, which were in agreement with Figure 3c.

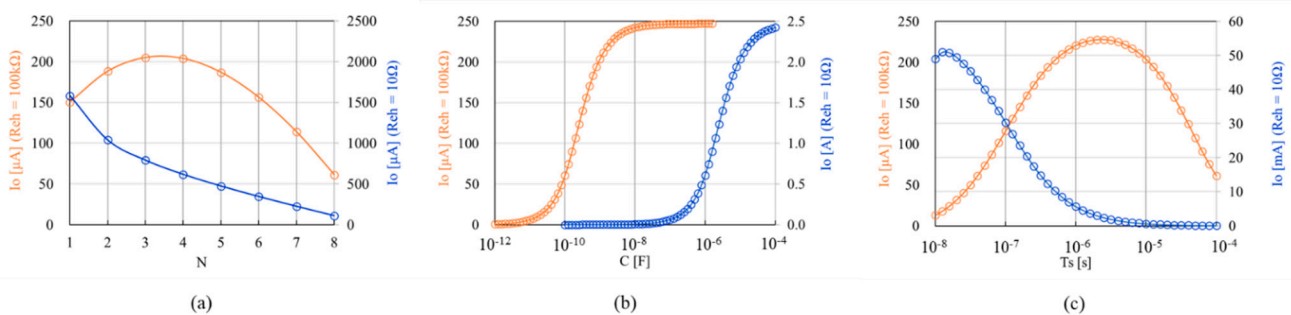

**Figure 3.** Io as a function of N (**a**); C (**b**); $T_S$ (**c**).

How can circuit designers maximize Io when the total capacitance is given? $I_O$ vs. $T_S$ plots for various N can tell them the answer. Figure 4 is a demonstration assuming

$C_{TOT} = CN = 100$ pF and the other parameters are given by Table 1. The optimum Ts to maximize Io depended on N because $\tau$ varied as $N^{-2}$ when CN was constant ($\tau = R_{ET}$ $C/N = R_{ET} C_{TOT}/N^2$). One can find the maximum Io ($I_{O\_MAX}$) for each N at $T_{S\_OPT}$ from Figure 4a. Figure 4b shows $I_{O\_MAX}$ vs. N when CN = 100 pF. In this demonstration, one can extract 160 µA at 1 V with N = 3 and Ts= 500 ns. The above procedure to determine the optimum N and Ts will be used in Section 3.

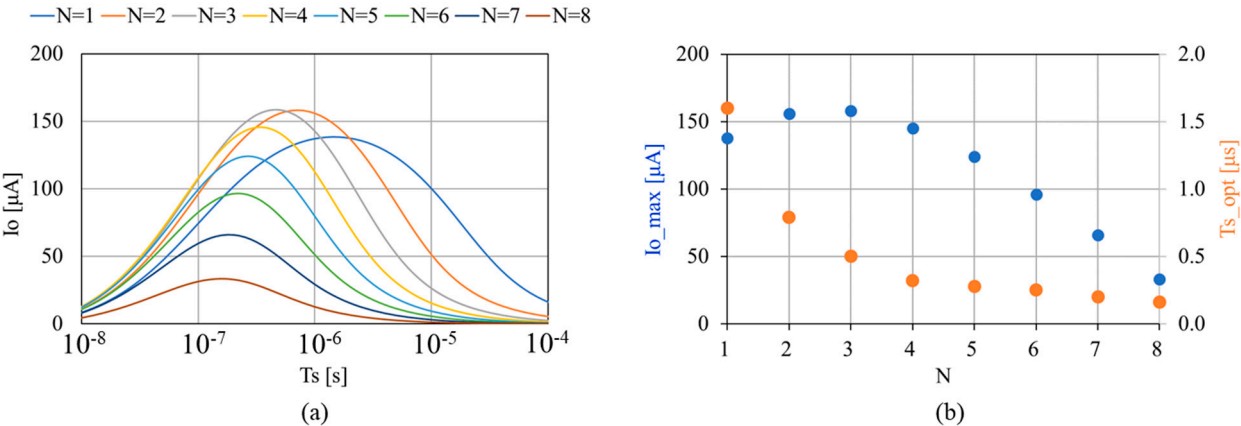

**Figure 4.** (**a**) Io as a function of Ts when $C_{TOT} = CN = 100$ pF; (**b**) $I_{O\_MAX}$ vs. N.

Figure 5 shows η vs. N based on (9) and (21). The two lines were identical. As N increased, the voltage ripple of each capacitor decreased, which contributed to a reduction in conduction loss, i.e., an increase in η [13]. Note that (9) and (21) did not take any parasitic capacitance into account for simplicity. When the parasitic capacitance, such as the top and bottom plate capacitance to the ground, and the junction capacitance of switches was considered, η was degraded especially for converters with many capacitors [14]. Improvement of the models discussed in this paper will be needed for more accurate initial design.

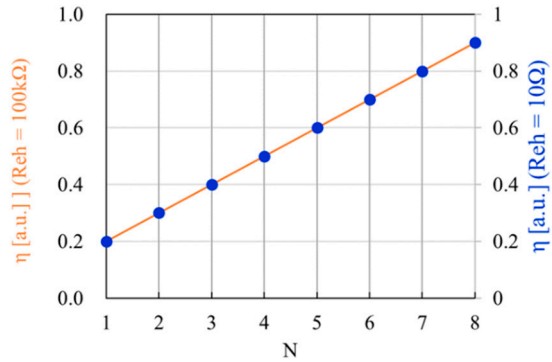

**Figure 5.** η as a function of N.

The above procedure could be performed for various $C_{TOT}$ as shown in Figure 6a. The maximum attainable power from a given ET specified with $V_{IN}$ and $R_{ET}$ ($P_{ATT}$) was given by (22c) under a power match, when the input impedance of the converter was matched with $R_{ET}$. In case of the conditions given in Table 1, $P_{ATT}$ was 250µW (10 $V^2/4 \times 100$ kΩ). As shown in Figure 6a, by increasing $C_{TOT}$, $I_{O\_MAX}$ approached $P_{ATT}/V_O$. The value of $N_{OPT}$ that provided the largest $I_{O\_MAX}$ was the one with a small $C_{TOT}$, whereas the value given by (22b) was one with a sufficiently large $C_{TOT}$.

One can draw Figure 6b by combining Figure 6a with Figure 5, which suggests that there was no chance to design an SC-VDC for highly resistive ET to maximize both $I_O$ and η. Which one should be prioritized for circuit designers? If a set of ET and SC-VDC was

considered as a power source, which was a viewpoint from the load, Io must be a higher priority than η.

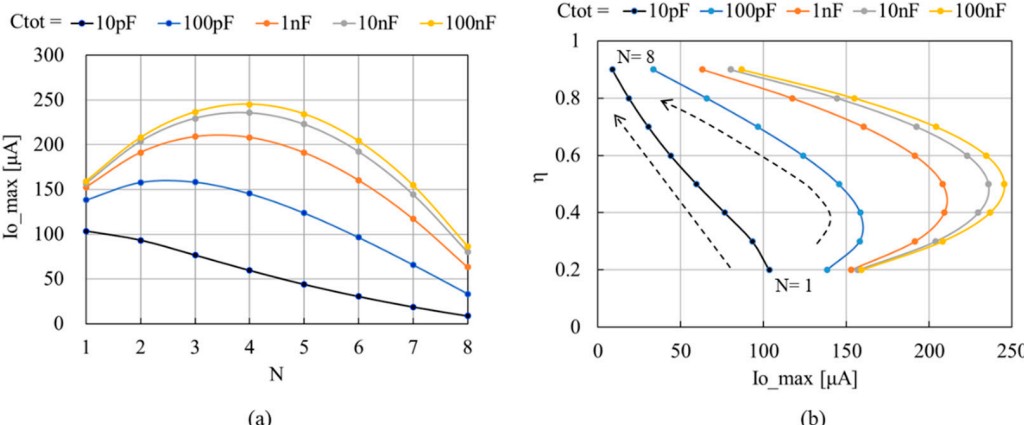

**Figure 6.** (**a**) $I_{O\_MAX}$ as a function of N for each value of $C_{TOT}$; (**b**) η as a function of $I_{O\_MAX}$ for each value of $C_{TOT}$.

Let us analyze Figure 6a in more detail. Figure 7a–c is $N_{OPT}$ (a), $T_{S\_OPT}$ (b) and $I_{O\_MAX}$ (c) as a function of $C_{TOT}$ based on Figure 6a. With $C_{TOT}$ of 10 pF, $N_{OPT}$ was one and $T_{S\_OPT}$ was 400 ns. This condition was close to a "no $R_{ET}$ case", which had $N_{OPT}$ of one. $I_{O\_MAX}$ was 100 µA at the most. Circuit designers may want to have a higher Io because the attainable output current was 250 µA. There was no other way to increase Io without a sacrifice of circuit area. Even though one could increase Io by increasing $C_{TOT}$, the rate of increase in Io was noticeably lower than the rate of increase in $C_{TOT}$. This was because Ts must also be increased for an increased τ, and $N_{OPT}$ must be increased as well. The larger the value of N, the smaller the series capacitance C/N, and therefore the lower Io. For instance, one can have Io of 200 µA with $C_{TOT}$ of 1 nF. With 100× $C_{TOT}$, Io barely increased by a factor of two.

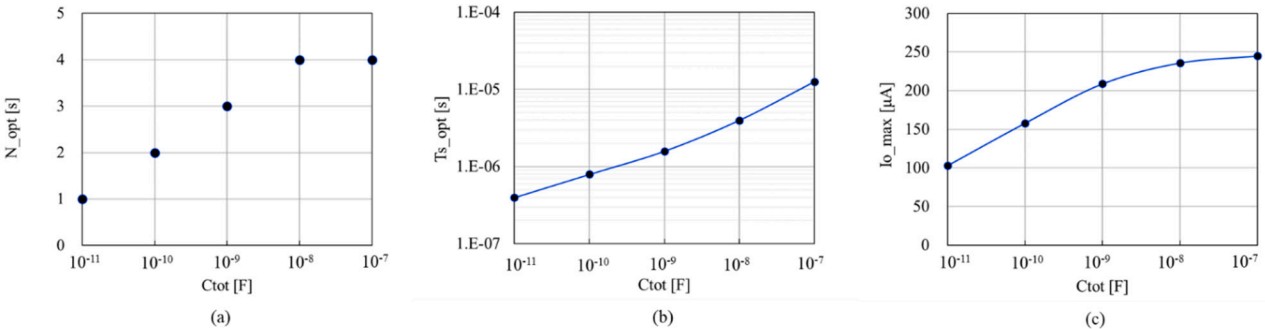

**Figure 7.** $N_{OPT}$ (**a**); $T_{S\_OPT}$ (**b**); $I_{O\_MAX}$ (**c**), as a function of $C_{TOT}$.

### 2.4. Validation of the Model

To validate the model expressed by (18), a two-stage SC-VDC was designed in 250 nm BCD technology, as shown in Figure 8a. First, 12 V CMOS transistors were used to manage $V_{IN}$ of 10 V in SPICE simulation. Transistors need to operate in a safe-operating region, i.e., the drain (source) voltage of N(P)MOSFETs must be equal to or greater than the source (drain) voltage. As a result, some switches were realized with two series transistors whose gates are driven by "ser1" and "ser2". The timings were slightly different, as shown in Figure 8b.

SPICE simulations were run with various Ts, resulting in Figure 9. The model still had a mismatch against SPICE, but if the model was requested to determine the optimum conditions for Ts (500 ns in this example), the model was considered to be in agreement

with the SPICE result. To design SC-VDCs precisely, one could run SPICE multiple times, starting at the conditions which the model predicts.

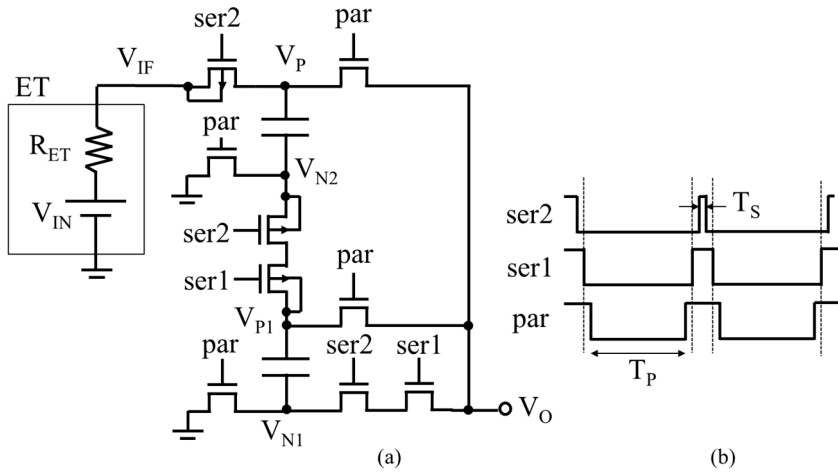

**Figure 8.** (**a**) Circuit diagram with N = 2 for model validation; (**b**) waveform of control signals.

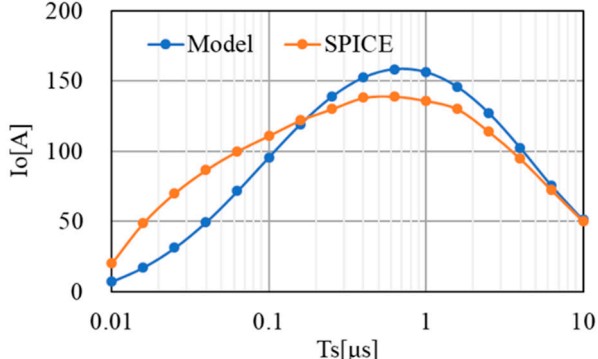

**Figure 9.** Comparison of Io vs. Ts between model calculation and SPICE simulation.

## 3. Optimum Design Flow

In this section, a design flow is proposed to design an SC-VDC for highly resistive ET so that the circuit area is minimized under the condition that a target Io ($I_{O\_TARGET}$) is provided at a target Vo. The key idea is (1) one can design an SC-VDC to maximize Io when its circuit area or the total capacitance is given; (2) if the maximized Io provided in (1) is lower (higher) than $I_{O\_TARGET}$, one can increase or decrease the circuit area gradually; and (3) finally one reaches SC-VDC with a minimum circuit area to barely output $I_{O\_TARGET}$.

Figure 10 shows the design flow. The following are performed step by step.

(S1) $V_{IN}$ and $R_{ET}$ are assumed from the ET side, and Vo and $I_{O\_TARGET}$ are required from the load side. At this point, one can check whether there is any solution for SC-VDC based on (22c). If the estimated $P_{ATT}$ is lower than Vo $\times$ $I_{O\_TARGET}$, the circuit designers must request to increase the input power to the ET side, or to decrease the output power to the load side.

(S2) An initial value of the total capacitance CN is assumed, which is called Ao in this flow. One can start at any value for Ao because the feedback loop will reach the final solution as long as there is a value for Ao.

(S3) One can draw Io vs. Ts for each SC-VDC with a different N by using (18), as shown in Figure 4a.

(S4) One can find a maximum $I_{O\_MAX}$ at an optimum $T_{S\_OPT}$ among all of the possible designs, as shown in Figure 4b.

(S5) Then, a parameter $\alpha$ is calculated with Io/$I_{O\_TARGET}$, which indicates how much Io deviates from the target.

(S6), (S8) If $1 < \alpha < 1.1$, the design determined in (S4) is the optimum design, which has a minimum circuit area with a design margin of 10% or less in Io. Therefore, the design flow is closed. If $\alpha < 1$, it is considered that Ao is not sufficient to output $I_{O\_TARGET}$ at Vo.

(S7) Then, Ao is increased by a factor of $1/\alpha$. The second trial starts at (S3) with Ao/$\alpha$.

(S8) If $\alpha > 1.1$, Ao is more than enough to output the $I_{O\_TARGET}$ at Vo. Then, Ao is decreased by a factor of Ao/$\alpha$ at (S7) to feedback to (S3) for the next loop.

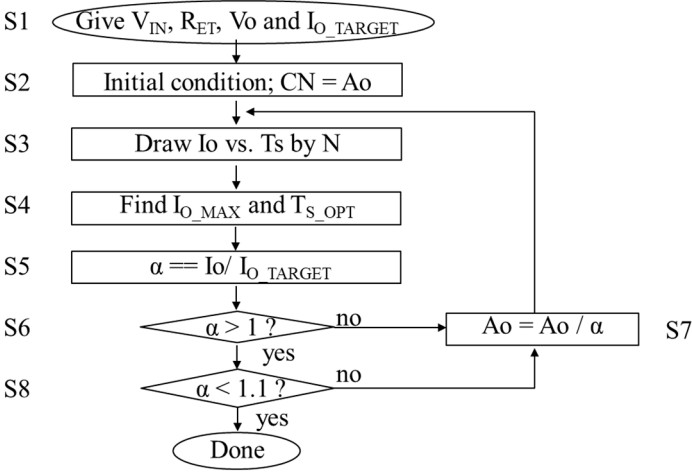

**Figure 10.** Optimum design flow to have a minimum circuit area.

A demonstration is made for $I_{O\_TARGET}$ of 100 µA at Vo of 1 V and $A_O$ of 100 pF. By running the design flow as shown in Figure 10, $C_{TOT}$, $T_{S\_OPT}$, $N_{OPT}$ and thereby $I_{O\_MAX}$ were determined in each loop as shown in Figure 11. $N_{OPT}$ varied from three in the first loop to one in the final tenth loop in this demonstration. $T_{S\_OPT}$ increased at the moment when $N_{OPT}$ decreased, because $\tau$ decreased. In total, Ao gradually decreased from an initial condition of 100 pF to the final value of 7.5 pF where $I_{O\_MAX}$ reached 110 µA.

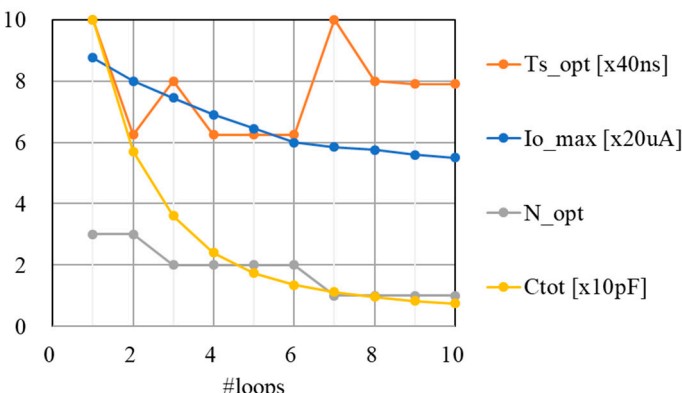

**Figure 11.** $I_{O\_MAX}$, $T_{S\_OPT}$, $N_{OPT}$ and $C_{TOT}$ vs. loop cycles under the condition of $I_{O\_TARGET}$ = 100 µA and Ao = 100 pF.

## 4. Conclusions

This paper developed a circuit model of DC-DC SC-VDC for highly resistive ET. Dependence on $T_{S\_OPT}$ and $N_{OPT}$ to maximize $I_O$ on $R_{ET}$, $V_{IN}$ and $V_O$ was demonstrated under the condition that a circuit area was given. Then, the optimum design flow was proposed to minimize the circuit area to meet the constraints of $R_{ET}$, $V_{IN}$, $V_O$ and $I_O$. Its demonstration was presented. The results in this paper will be used to design AC-DC SC-VDC for highly resistive ET effectively.

**Author Contributions:** Conceptualization, T.T.; methodology, Y.D. and T.T.; software, Y.D.; validation, Y.D. and T.T.; formal analysis, Y.D. and T.T.; investigation, Y.D. and T.T.; writing—original draft preparation, Y.D.; writing—review and editing, T.T.; funding acquisition, T.T. All authors have read and agreed to the published version of the manuscript.

**Funding:** This research received no external funding.

**Acknowledgments:** This work was supported by d-lab.VDEC, Synopsys, Inc. (Mountain View, CA, USA) and Cadence Design Systems, Inc. ( San Jose, CA, USA).

**Conflicts of Interest:** The authors declare no conflict of interest.

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
