# Peer review of "Design of Switched-Capacitor DC-DC Voltage-Down Converters Driven by Highly Resistive Energy Transducer"

_electronics, doi:10.3390/electronics11121874_

Round 1
Reviewer 1 Report
In this concept paper, to develop switched-capacitor voltage-down converters (SC-VDCs) for electrostatic vibration energy transducers ideally, the authors first design DC-DC SC-VDCs with high RET, then design AC-DC SC-VDCs based on the 1st step. Their paper addresses phase 1 circuit analysis and design optimization. To minimize circuit space, switching frequency, number of stages, and stage capacitance are dictated by RET, VIN, load voltage (Vo), and current (Io). Their design is simulated using 250nm BCD. An optimal design flow is suggested for DC-DC SC-VDCs with high RET, VIN, Vo, and Io. This paper is suited to be published with some minor comments that the authors should consider:
1. In line 43: "To the best knowledge of the authors, there has no formulation to design switched-capacitor voltage-up converters for highly resistive energy transducer..." The paper's topic is voltage-down converters. Please clarify the statement if this is correct. Or is this statement related to the authors' research problem?
2. It is good to compare the techniques related to dc-dc converters such as reflected in the following but not limited to these references:
a. L. Lai, et al. "Monolithic integrated high frequency GaN DC-DC buck converters with high power density controlled by current mode logic level signal." Electronics 9.9 (2020): 1540.
b. C.-C. Wang, et al. "A 40-nm CMOS Piezoelectric Energy Harvesting IC for Wearable Biomedical Applications." Electronics 10.6 (2021): 649.
Add more references from 2017 up to present which are related and comparable to this study.
Reviewer 2 Report
This paper presents an optimal design process for designing high RET drive DC-DC SC-VDC with minimum circuit area under given RET, VIN, Vo and Io conditions. The relationship between variables is obtained from a set of equations and verified by simulation with control variables.
However, this paper has some deficiencies as follows:
1. In the abstract, it is not clear to propose an optimum design of AC-DC SC-VDCs, there is no relevant introduction in this paper.
2. Lines195, 213, 233, what does the word SW-VDC mean?
3. Equation (2): Replace the output charge in the parallel state ( QOP ) with the output charge in the serial state( QOS ).
4. It is not clear how to minimize circuit area by an optimal design.
